# A Pilot Study on Efficacy of Lipid Bubbles for Theranostics in Dogs with Tumors

**DOI:** 10.3390/cancers12092423

**Published:** 2020-08-26

**Authors:** Inoru Yokoe, Yusuke Murahata, Kazuki Harada, Yuji Sunden, Daiki Omata, Johan Unga, Ryo Suzuki, Kazuo Maruyama, Yoshiharu Okamoto, Tomohiro Osaki

**Affiliations:** 1Joint Department of Veterinary Clinical Medicine, Faculty of Agriculture, Tottori University, 4-101 Koyama-Minami, Tottori 680-8553, Japan; inoru.y@gmail.com (I.Y.); ymurahata@muses.tottori-u.ac.jp (Y.M.); k-harada@tottori-u.ac.jp (K.H.); sunden@tottori-u.ac.jp (Y.S.); yokamoto@muses.tottori-u.ac.jp (Y.O.); 2Laboratory of Drug and Gene Delivery Research, Faculty of Pharma-Science, Teikyo University, 2-11-1 Kaga, Itabashi-ku, Tokyo 173-8605, Japan; omata@pharm.teikyo-u.ac.jp (D.O.); johan.unga@gmail.com (J.U.); r-suzuki@pharm.teikyo-u.ac.jp (R.S.); 3Laboratory of Theranostics, Faculty of Pharma-Science, Teikyo University, 2-11-1 Kaga, Itabashi-ku, Tokyo 173-8605, Japan; maruyama@pharm.teikyo-u.ac.jp

**Keywords:** contrast-enhanced ultrasonography, dog, lipid bubbles, tumor, theranostics, ultrasound

## Abstract

The combined administration of microbubbles and ultrasound (US) is a promising strategy for theranostics, i.e., a combination of therapeutics and diagnostics. Lipid bubbles (LBs), which are experimental theranostic microbubbles, have demonstrated efficacy in vitro and in vivo for both contrast imaging and drug delivery in combination with US irradiation. To evaluate the clinical efficacy of LBs in combination with US in large animals, we performed a series of experiments, including clinical studies in dogs. First, contrast-enhanced ultrasonography using LBs (LB-CEUS) was performed on the livers of six healthy Beagles. The hepatic portal vein and liver tissue were enhanced; no adverse reactions were observed. Second, LB-CEUS was applied clinically to 21 dogs with focal liver lesions. The sensitivity and specificity were 100.0% and 83.3%, respectively. These results suggested that LB-CEUS could be used safely for diagnosis, with high accuracy. Finally, LBs were administered in combination with therapeutic US to three dogs with an anatomically unresectable solid tumor in the perianal and cervical region to determine the enhancement of the chemotherapeutic effect of liposomal doxorubicin; a notable reduction in tumor volume was observed. These findings indicate that LBs have potential for both therapeutic and diagnostic applications in dogs in combination with US irradiation.

## 1. Introduction

“Theranostics”, a portmanteau of therapeutics and diagnostics, refers to a concept or system in which treatment and diagnosis are performed simultaneously using a single medical device and drug [1,2]. Recent advanced imaging modalities underlie the rise of this concept; ultrasound (US) in particular, is a promising type of mechanical energy for use in theranostics due to its non-invasive and deep-reaching characteristics [3]. Microbubbles (MBs) are used widely as a US contrast agent in contrast-enhanced ultrasonography (CEUS) and are also considered a key agent for application in theranostics using US [3,4,5]. This is because the combination of MBs and US irradiation has demonstrated potential as a novel drug delivery system [3,4,5]. Microbubbles generally consist of an encapsulating shell (made of proteins, lipids, or polymers) and inner gas core (e.g., air, sulfur hexafluoride, and perfluorocarbons). The mean diameter of one MB particle is small enough (1–10 μm) to pass through capillary vessels and circulate throughout the body [4,5]. When MBs are exposed to low-pressure US irradiation, they commonly oscillate in response to the acoustic waves [6] and scatter acoustic signals with high frequencies [3,7]. An ultrasonic diagnostic device intercepts the scattered signals and then generates contrast-enhanced images [3,7]. Conversely, drug delivery using MBs is triggered by US irradiation at a relatively higher pressure. Under these circumstances, large cycles of expansion and contraction can lead to inertial cavitation and, ultimately, to the destruction of MBs [6,8]. This phenomenon generates reversible nano-pores in the biomembrane, through which substances such as drugs and DNA can subsequently pass [3,4,5]. Drug delivery using MBs and US is considered as an interesting approach to improve the delivery of therapeutic agents, especially to the brain and tumor tissues, which have physical and functional barriers. Thus, the use of MBs in combination with US provides a favorable theranostic technique.

Currently, three commercial MBs are commonly used for the diagnosis of focal liver lesions, namely, SonoVue, Definity, and Sonazoid [9]. The high diagnostic accuracy of liver CEUS using MBs has been reported in human medicine [10]. On the contrary, the therapeutic application of MBs in combination with US has not been established although a large number of in vitro and in vivo studies on drug delivery with MBs have been conducted [11,12]. A reason for this issue is that the commercial MBs are optimally designed for CEUS and are not considered for application in therapeutics or theranostics. Therefore, various experimental MBs have been designed and developed for use with US in theranostics [3,5]. Lipid bubbles (LBs) are an experimental MB preparation, with an average particle size of approximately 2.7 μm, developed for both therapeutic and diagnostic applications [13,14,15]. The lipid monolayer structure of LB particles comprises phospholipids modified with polyethyleneglycol of molecular weight 2000 (PEG2000) and contains perfluoropropane gas inside. In a previous study, it was demonstrated that LBs have US contrast effects both in vitro and in vivo [16]. Additionally, in tumor-bearing mice, it was reported that the combination of LBs and US irradiation induced the delivery of conventional doxorubicin into tumor tissue, resulting in the enhancement of the anti-tumor effect of doxorubicin [17]. Thus, LBs are a promising theranostic preparation due to their contrast and drug-delivery effects with US irradiation. However, the effects of LBs have been confirmed only in mice; therefore, it is necessary to elucidate their efficacy in larger animals such as dogs if they are to be developed for clinical use in humans. The aim of the present translational research was to evaluate the short-term safety and US contrast effect of LBs in healthy dogs and to determine the clinical efficacy of the combination of LBs and US for the diagnosis and treatment of spontaneously arising tumors in dogs.

## 2. Results

### 2.1. US Contrast Effect and Safety of LBs in Healthy Beagles

Hematological and biochemical changes in the administration of LBs are presented in Table 1. All hematological and biochemical variables before and after the administration of LBs were within reference interval for dogs. In hematological tests, the pre-injection and post-injection platelet values were significantly different (37.9 vs. 30.7 × 10^4^/μL, *p* = 0.020). In biochemical tests, the pre-injection and post-injection blood urea nitrogen and creatinine values were significantly different (15.2 vs. 11.5 mg/dL, *p* = 0.0005, 0.72 vs. 0.67 mg/dL, *p* = 0.034, respectively). There were no significant differences in other parameters before and after injection. The symptoms of mucosal hyperemia, facial swelling, anorexia, hypoactivity, and collapse were not observed around and 24 h after the administration of LBs in all the Beagles. Figure 1 shows images for CEUS using LBs (LB-CEUS) and CEUS using Sonazoid (Sonazoid-CEUS). Following the administration of LBs, thin hepatic arteries were enhanced, and the enhancement was diffused rapidly. Subsequently, the hepatic portal vein and liver tissue were enhanced within 10–20 s. Enhancement of the hepatic portal vein appeared strongest around 1 min (Figure 1B), after which it gradually faded (Figure 1C) and disappeared (Figure 1D). Liver tissue remained enhanced throughout the observation period of 15 min (Figure 1D). The enhancement of the liver tissue in Sonazoid-CEUS was stronger than that in LB-CEUS (Figure 1F,G). Figure 2 shows the time-mean grayscale intensity (MGI) curve of the hepatic portal vein and liver tissue. The MGI of the hepatic portal vein peaked 0.5 min after the administration of LBs, and then gradually decreased; the MGI values returned to baseline values at 8 min. The MGI of liver tissue peaked 1 min after the administration of LBs. After peaking, the MGI of liver tissue was maintained at the same level throughout the observation period. The time-MGI curves of the hepatic portal vein and liver tissue were similar between LB-CEUS and Sonazoid-CEUS. In Sonazoid-CEUS, the MGI of the hepatic portal vein peaked 0.5 min after the administration of Sonazoid, and then gradually decreased toward the baseline. The MGI of liver tissue peaked 0.5 min after the administration of Sonzaoid. The MGI of liver tissue even maintained at high level in 15 min although it gradually decreased after peaking. The peak MGI values in the hepatic portal vein did not differ significantly between LB-CEUS and Sonazoid-CEUS (154.0 vs. 168.7, *p* = 0.404). Meanwhile, the peak MGI value of liver tissue was significantly lower with LB-CEUS than with Sonazoid-CEUS (67.2 at 1 min vs. 102.5 at 0.5 min, *p* = 0.010). The MGI value of liver tissue at 15 min did not differ significantly between LB-CEUS and Sonazoid-CEUS (58.9 vs. 48.0, *p* = 0.116). The timings of the three phases for LB-CEUS were set as follows: arterial phase—approximately from 0 to 30 s; portal phase—approximately from 30 s to 2 min; and Kupffer phase—approximately from 10 min after administration.

### 2.2. CEUS in Dogs with Focal Liver Lesions

Table 2 presents the summary of the results of the 21 dogs included in this study. The ages of the 21 dogs ranged from 8.5 to 14.3 years old (median, 11.7 years old). Their body weights ranged from 1.8 to 23.2 kg (median, 8.9 kg). Their breeds were different; Mix (*n* = 4), Miniature Dachshund (*n* = 4), Chihuahua (*n* = 3), Miniature Schnauzer (*n* = 2), Beagle (*n* = 2), Welsh Corgi (*n* = 1), Boston Terrier (*n* = 1), Golden Retriever (*n* = 1), Yorkshire Terrier (*n* = 1), French Bulldog (*n* = 1), and Jack Russell Terrier (*n* = 1). Of the 21 dogs with focal liver lesions, 6 dogs had benign lesions and 15 had malignant tumors. Diagnosis of the lesions of the 21 dogs are presented in Table 2. Six dogs with benign lesions were histopathologically diagnosed. Eleven of 15 dogs with a malignant tumor had a primary lesion. The remaining four dogs had a metastatic lesion. Fourteen malignant tumors, except for one malignant melanoma, were diagnosed through histopathological examination; the malignant melanoma was diagnosed through cytological examination.

In clinical cases, LB-CEUS was able to visualize all three phases: arterial phase, portal phase, and Kupffer phase. The symptoms of mucosal hyperemia and facial swelling related to the administration of LBs were not observed in any of the 21 dogs. The findings of each phase in LB-CEUS and Sonazoid-CEUS were consistent in 13 dogs. The findings of LB-CEUS are summarized in Table 2. All 15 malignant tumors presented a complete enhancement defect (CD) or an irregular enhancement defect (ID) in the Kupffer phase. The findings of CD and ID in the Kupffer phase were significantly related to malignancy, with a sensitivity of 100.0%, specificity of 83.3%, positive predictive value of 93.8%, and negative predictive value of 100.0% (*p* = 0.0003). Conversely, five of the six benign lesions showed no enhancement defect (ND) in the Kupffer phase. One cholangiocellular adenoma showed CD. The finding of ND in the Kupffer phase was significantly related to benignancy, with a sensitivity of 83.3%, specificity of 100.0%, positive predictive value of 100.0%, and negative predictive value of 93.8% (*p* = 0.0003). In the portal phase, the finding of isoenhancing was significantly related to benignancy, with a sensitivity of 83.3%, specificity of 80.0%, positive predictive value of 62.5%, and negative predictive value of 92.3% (*p* = 0.014). In the arterial phase, no significant findings were found in relation to benignancy or malignancy. Figure 3 and Figure 4 are representative images of LB-CEUS and Sonazoid-CEUS in the clinical study; these are described in detail in the respective legends.

### 2.3. Anti-Tumor Effects of LBs in Combination with Therapeutic US and Liposomal Doxorubicin

The dogs used in the clinical study are described in Table 3 (Appendix A). The tumor types displayed by the cases were hemangiopericytoma, ceruminous gland adenocarcinoma, and thyroid carcinoma (suspected). The case No. 1 dog could not receive a surgical treatment because the treatment could probably cause fecal incontinence by excision of the external anal sphincter. Similarly, a surgical treatment was avoided for case No. 2 and 3 due to the anatomical tumor location and the tumor fixation to the carotid artery, respectively. Only case No. 3 received toceranib simultaneously during the observation period. Treatment was administered two, six, and four times to cases No. 1, 2, and 3, respectively. The tumor mass volume decreased markedly following treatment, to 28.9, 28.2, and 44.3% in cases No. 1, 2, and 3, respectively (Figure 5). The survival times were 52 and 127 days in cases No. 1 and 2, respectively (Appendix A). Cases No. 1 and 2 died of a cause unrelated to the tumors. Case No. 3 was lost to follow-up by day 83, and its outcome was unknown (Appendix A). Detailed information on the treatments received by each dog is presented in Table 4. In all cases, LB-CEUS was performed on tumor tissue before and after treatment. In all tumor tissues, a hypoenhancing area was observed through LB-CEUS after treatment.

Figure 6 shows the LB-CEUS images of case No. 1 at the first treatment on day 0. The tumor tissue was enhanced by the administration of LBs. Simultaneously performed LB-CEUS showed that the enhancement of the tumor tissue was completely eliminated by therapeutic US irradiation (Figure 6E–H). Figure 7 presents angiographic computed tomography (CT) and LB-CEUS images of case No. 2. In this dog, CT images revealed a reduction in tumor size following the third treatment. Figure 8 shows angiographic CT images and LB-CEUS images of case No. 3. In this case, a CT examination performed on day 0 revealed that the dog had pulmonary metastases. Following a series of treatments, the size of the primary tumor in the neck decreased, while there were no changes in pulmonary metastases. LB-CEUS after the first treatment generated a hypoenhancing area at the center of the lesion, which was not observed in LB-CEUS before the treatment. The detailed explanation of each figure is provided in the respective legend.

## 3. Discussion

LBs have been developed for use in both diagnosis and treatment. Although previous studies have demonstrated the contrast and drug delivery effects of LBs with US in vitro and in vivo [13,14,15,16,17], non-clinical research in large animals is essential to elucidate their safety and clinical profiles. In humans, MB preparations have mainly been used for diagnosing liver tumors [7]. Therefore, we applied LBs to healthy dogs in order to evaluate their safety and US contrast effects in the liver. In the present study on Beagle, LBs were administered intravenously at a dose of 0.015 mL/kg (2.5 × 10^9^ particles/mL). The values of platelet, blood urea nitrogen, and creatinine before injection and 24 h after injection were significantly different; however, these differences were very slight. All hematological and biochemical variables before and after the administration of LBs were within reference interval for dogs. In addition, the acute allergic symptoms of mucosal hyperemia, facial swelling, anorexia, hypoactivity, and collapse were not observed with the administration of LBs. These results indicated that single injection of LBs at the abovementioned dose had a low risk to induce hematological changes, hepatic and nephrotic disorders, and allergic reactions in healthy dogs. In human medicine, MB preparations are considered to have a highly favorable safety profile [7,9]. MB preparations have no cardiac, hepatic, and nephrotic toxic effects. Although anaphylactoid reactions are the prevalent adverse effect of MBs, the probability of their occurrence is estimated to have a rate of 0.014%, which is lower than that of adverse effects of X-ray contrast agent [7]. In preclinical studies of healthy dogs, the good safeness of mainstream MB preparations (SonoVue and Sonazoid) was confirmed with no adverse reactions including allergic symptoms and clinical pathological changes [18,19]. Our results from the Beagle study suggested that LBs, as well as the two MBs, might be safe for use in dogs. Regarding the US contrast effect, the hepatic portal vein and liver tissue were both enhanced with the administration of LBs. Notably, the enhancement of liver tissue was observed throughout the observation period in both LB-CEUS and Sonazoid-CEUS. The timing and the MGI value of the peak enhancement for liver tissue were equal between the two CEUS methods, while the peak enhancement for liver tissue was lower with LB-CEUS. As a result, the time-MGI curves of the two CEUS methods were approximately similar. These results indicated that LB-CEUS could visualize the three phases of the arterial, portal, and Kupffer phase in the same manner as Sonazoid-CEUS. Therefore, the timings of the three phases were set for LB-CEUS based on the results in the Beagle study. Sonazoid is unique in its ability to be phagocytosed by Kupffer cells in liver sinuses [20,21]. Because of this, Kupffer phase images can be depicted with liver tissue enhancement in Sonazoid-CEUS [19,20,22,23,24,25]. We considered LBs, as well as Sonazoid, to be phagocytosed by Kupffer cells and to provide the liver tissue enhancement. Conversely, the mechanism of liver tissue enhancement in LB-CEUS was unclear in the present study. A previous report showed that liposomes modified with PEG tend to avoid phagocytosis by Kupffer cells in vitro and in vivo [26]. The phospholipid shell of LBs is modified with PEG 2000; therefore, our results might be inconsistent with those of the previous study. Further studies are needed to elucidate the mechanism through which LBs induce contrast effects in liver tissue. Nevertheless, it was revealed that LB-CEUS had potential to evaluate liver lesions with the three phases as well as Sonazoid-CEUS.

LBs were found to have a favorable safety profile and a US contrast effect for liver tissue in healthy dogs as described above. Next, we conducted a clinical study of LB-CEUS in dogs with focal liver lesions to evaluate their diagnostic ability. In clinical cases, the arterial phase, portal phase, and Kupffer phase were visualized through LB-CEUS. The contrast findings of each CEUS method were consistent in the 13 dogs that underwent LB-CEUS and Sonazoid-CEUS. These results implied that LB-CEUS was able to provide images equivalent to those obtained by Sonazoid-CEUS. We subsequently added eight cases and performed LB-CEUS on a total of 21 dogs. Following LB-CEUS in 21 dogs, a defect in contrast enhancement in the Kupffer phase was considered the most important finding supporting the diagnosis of liver lesions. All 15 malignant tumors presented CD or ID, and five of the six benign lesions showed ND in the Kupffer phase. ID and CD findings were significantly related to malignancy, and ND findings were significantly related to benignancy with high sensitivity, specificity, positive predictive value, and negative predictive value. These results of our study were consistent with those in previous reports regarding Sonazoid-CEUS in dogs [24,25] and in humans [27], which demonstrated that enhancement defects in the Kupffer phase were the most important findings for Sonazoid-CEUS. Kupffer cells exist in benign liver lesions, while they are lacking in malignant hepatic tumors in both dogs and humans. Because of this, benign lesions are imaged with no enhancement defect, and malignant tumors are depicted as a contrast defect area through Sonazoid-CEUS due to phagocytosis by Kupffer cells [22,24,25,27]. In the present study, LB-CEUS might demonstrate the high diagnostic ability for hepatic focal lesions due to the depiction of Kupffer phase images. Conversely, only one cholangiocellular adenoma showed the CD finding mimicking malignancy in the Kupffer phase. This result was consistent with that of a recent study, in which three canine cholangiocellular adenomas presented enhancement defects in the Kupffer phase with Sonazoid-CEUS [28]. In dogs, colangiocellular adenoma has a cystic structure in which locules are lined by a low cuboidal epithelium [29]. Hence, the CD finding for cholangiocellular adenoma in this study might reflect the absence of Kupffer cells in the lesion. Regarding safety, the allergic symptoms of mucosal hyperemia and facial swelling were not observed following the administration of LBs among the 21 dogs. This result indicated that the administration of LBs had a low risk for acute anaphylactoid reactions in dogs with naturally-occurring focal liver lesions. The results of the present study suggested that LB-CEUS, as well as Sonazoid-CEUS, could be performed safely and be useful for diagnosing focal liver lesions in dogs.

Finally, we conducted an exploratory clinical study to investigate the anti-tumor effects of LBs, therapeutic US, and Doxil, a liposomal formulation of doxorubicin. We hypothesized that the combination of LBs and US irradiation enhanced the anti-tumor effects of Doxil. In Doxil, which has a mean particle size of approximately 100 nm, doxorubicin is encapsulated in a PEGylated phospholipid bilayer [30]. Due to these structural features, Doxil exhibits a prolonged circulation time in dogs [31] and in humans [32], and accumulates in human tumor tissue [32]. However, the clinical efficacy of liposomal doxorubicin in veterinary medicine has been reported to be limited [33]. In a previous study investigating the use of liposomal doxorubicin in 51 dogs with various malignant tumors, the tumor volume was reduced by more than 50% in only 13 dogs (25.5%) [34]. Another clinical study involving 14 dogs with splenic hemangiosarcoma reported that clinical outcomes following intraperitoneal injection of liposomal doxorubicin were not superior to those following an intravenous injection of conventional doxorubicin [35]. In contrast, in the present study, although the types and stages of tumors differed, more than 50% reduction in tumor volume was observed in all three dogs. Notably, in case No. 3, in which thyroid carcinoma was suspected, only the primary tumor tissue in the cervical region treated with LBs and therapeutic US decreased in size, while there were no changes in lung metastases. These results indicated that the combination of LBs and US might improve the anti-tumor effects of Doxil in tumor tissues treated with US irradiation. In human medicine, an epoch-making attempt has been made to enhance the effects of gemcitabine with MBs and US [36]. This was a phase 1 clinical trial that combined SonoVue, diagnostic US, and gemcitabine in patients with inoperable pancreatic cancer. The study reported positive results, including a reduction in tumor diameter in 5 of 10 patients and an increase in the number of treatment cycles without any notable adverse events. The authors speculated that the therapeutic efficacy was due to the potential increase of gemcitabine uptake by US in combination with SonoVue. Similarly, we assumed that the combination of LBs and US irradiation may promote the delivery of Doxil to tumor tissue. In the present study, it was revealed that therapeutic US irradiation completely eliminated the enhancement in the tumor tissue. This result demonstrated that the therapeutic US could cause inertial cavitation and destruction of LBs in tumor tissues. Therefore, we considered that inertial cavitation and destruction of LBs caused by US irradiation might improve the permeability of tumor vasculature. Physical interactions between acoustically excited LBs and capillary walls could create nano-pores in endothelial cells or open gaps in endothelial cells, resulting in the delivery of Doxil into tumor tissue. Furthermore, it was reported that microvascular rupture could occur when MBs were irradiated by US in vivo [37]. In the present study, the contrast enhancement of tumor tissue in LB-CEUS decreased after the combination treatment. If capillary walls are ruptured in response to LBs and US irradiation, microthrombosis might occur in tumor vasculature, and low enhancement of tumor tissue would be seen in CEUS. In that case, Doxil might extravasate through the disrupted walls and provide a greater anti-tumor effect. Nevertheless, further in vitro and in vivo studies are needed to elucidate the underlying mechanism of action.

There are several limitations in the present study. Regarding the safety of LBs, the simple safety assessment with a single dose, a single shot, and limited laboratory tests was performed using Beagles. In clinical use, there is a possibility that LBs are administered repeatedly in a short period or at much a higher dose. Therefore, repeated dose tests and higher dose tests in healthy dogs are necessary to evaluate detailed safety profile. In addition, in the clinical study for liver CEUS, dogs were monitored around the injection of LBs. The monitoring was not sufficient to evaluate subacute and delayed adverse reactions. Another clinical study with a follow-up survey is needed to evaluate the occurrence of subacute and delayed adverse events related to LB-CEUS. Regarding the diagnostic ability of liver LB-CEUS in dogs, referring to each CEUS finding was not suitable because the present clinical study had a small number of dogs in each disease type. To evaluate characteristic LB-CEUS findings in each type of lesion, a further clinical study with a larger population is necessary. Regarding the therapeutic application of LBs, the present study of the combination treatment was just exploratorily performed. The study included only three dogs with different tumor types, and had no control group. In addition, the study lacked a proper safety assessment. Therefore, in the future, further prospective clinical studies with a large number of cases with specific tumor types are required to evaluate the safety and treatment efficacy of a combination of LBs, therapeutic US, and Doxil.

## 4. Materials and Methods

### 4.1. Preparation of LBs

#### 4.1.1. Reagents

1,2-distearoyl-sn-glycero-3-phosphocholine (DSPC), 1,2-distearoyl-sn-glycero-3-phosphoglycerol (DSPG), and N-(carbonyl-methoxypolyethyleneglycol 2000)-1,2-distearoyl-sn-glycero-3-phosphoethanolamine (DSPE-PEG2000) were purchased from NOF Corporation (Tokyo, Japan). Perfluoropropane was purchased from Takachiho Chemical Industrial Co., Ltd., (Tokyo, Japan).

#### 4.1.2. Preparation of Liposomes and LBs

As previously reported [15], LBs were prepared by homogenization of a lipid dispersion in a perfluoropropane atmosphere and then freeze dried. Liposomes composed of DSPC, DSPG, and DSPE-PEG2000 at a molar ratio of 30:60:10 were prepared using the lipid film hydration method. In brief, 300 µmol of all lipids were mixed with chloroform, methanol, aqueous ammonium solution, and MilliQ water (65:35:4:4, *v*/*v*/*v*/*v*). The lipid film was prepared using a rotary evaporator, followed by drying overnight in a vacuum desiccator to completely remove the solvents. Then, the lipid film was hydrated with 75 mL of 100 mM phosphate buffer (pH 7.4) for 30 min at 65 °C with shaking. The resulting liposomes were treated in a bath type sonicator (Bransonic 2150j-DTH, Branson Ultrasound Co., Danbury, CT, USA) for 10 min. A homogenizer (Labolution MarkII 2.5, Primix Corporation, Hyogo, Japan) was inserted into a mixing vessel containing 75 mL of liposomes and 225 mL of 100 mM phosphate buffer. The air in the vessel was replaced with perfluoropropane. The liposomes were homogenized at 7500 rpm for 60 min at 40 °C. The microbubbles dispersion was mixed with 18% sucrose solution at a 1:1 (*v*/*v*) ratio, and 2 mL of the mixture was dispensed into a 5 mL vial. The air in the head space was replaced with perfluoropropane. The vials were closed with rubber lids, followed by freezing at −30 °C. After freezing and opening the rubber lid, freeze-drying was performed at −30 °C for 1 h, −20 °C for 72 h, and 20 °C for 48 h, using a shelf temperature controlled drying chamber (Eyela FDU-1100 and Eyela DRC-1100, Tokyo Rikakikai Co. Ltd., Tokyo, Japan) filled with perfluoropropane. The vials were then closed with rubber lids and capped with an aluminum cap. Before administration to dogs, the freeze-dried LBs were reconstituted in 2 mL of distilled water and shaken gently.

### 4.2. US Contrast Effect and Safety of LBs in Healthy Dogs

#### 4.2.1. Animals

From the viewpoint of animal welfare, the animal protocols were prepared based on 3Rs (replacement, reduction and refinement) principles. The study was designed as a crossover study to minimize the number of animals included. Six adult healthy Beagles aged 2.9–11.1 years old (median, 4.6 years old), and weighing 11.0–12.6 kg (median, 12.2 kg) were used in this study. The health of the dogs was confirmed by general physical examination, and biochemical and hematological tests. No focal liver lesions were detected by conventional ultrasonography in any of the dogs. All animal protocols and procedures involving healthy animals were approved by the Animal Research Committee of Tottori University (project number: 15-T-41).

#### 4.2.2. CEUS for Liver and Laboratory Test

A US scanner (Arietta60, Hitachi Aloka Medical, Ltd., Tokyo, Japan) with a 1–6 MHz microconvex probe was used. In contrast harmonic imaging mode, the mechanical index was set at 0.18 and the background gain was set at 55 dB. Each dog was placed in dorsal recumbency without sedation and their cranial abdomen was shaved. LBs were administered to the dogs intravenously at a dose of 0.015 mL/kg (2.5 × 10^9^ particles/mL) through a 22-G catheter, followed by 3 mL heparinized saline injection. The probe was set close behind the xiphoid region in order to scan the liver and portal vein and get a single image. The scanning video started recording just before drug administration and ended at 15 min after drug administration. The dogs were monitored for any acute allergic symptoms occurring around the point of administration of LBs. Biochemical and hematological tests were performed 24 h after the administration of LBs. Complete blood count parameters were measured using an automated hematology analyzer (pocH-100iV Diff, Sysmex Co., Kobe, Japan). Plasma C-reactive protein concentration was measured with a canine CRP measurement kit (Laser CRP-2, Arrows Co. Ltd., Osaka, Japan). Blood urea nitrogen, creatinine, alanine aminotransferase, alkaline phosphatase, and albumin were assayed with a dry chemistry analyzer (FUJI DRI-CHEM 7000V, FUJIFILM Corporation, Tokyo, Japan). After a 7-day washout period, we performed Sonazoid-CEUS at a dose of 0.015 mL/kg (0.85 × 10^9^ particles/mL) in the same way. Sonazoid (Daiichi-Sankyo, Tokyo, Japan) was administered for comparison of the US contrast effect of LBs.

#### 4.2.3. Imaging Analysis

For quantitative analysis, individual US images were acquired 0, 0.2, 0.5, 1, 2, 3, 4, 5, 6, 7, 8, 9, 10, 12, and 15 min after the administration of LBs and Sonazoid by taking screenshots from the video. All images were analyzed using the image analysis software (ImageJ, US National Institutes of Health, http://rsb.info.nih.gov/ij/). In the software, a region of interest was set on the hepatic portal vein and liver tissue. The MGI of each region of interest was calculated within a range of 0 to 255.

### 4.3. CEUS in Dogs Bearing Focal Liver Lesions

#### 4.3.1. Animals

Twenty-one owned dogs were included in this clinical study. The dogs were brought to the Tottori University Veterinary Medical Center between June 2016 and August 2019. It was confirmed that they had one or more naturally-occurring focal liver lesions through ultrasonography. Each focal liver lesion was diagnosed by histopathological or cytological examination. The detailed information on the dogs is provided in the results Section 2.2 and in Table 2. The clinical protocol for the owned dogs, as well as the procedures they underwent, was approved by the Ethics Committee at the Faculty of Agriculture, Tottori University (ethical approval number: H28-001)

#### 4.3.2. Characterization of Focal Liver Lesions with Conventional Ultrasonography

Conventional ultrasonography was performed to characterize each lesion using one of the two US scanners, Arietta60 or HI VISION Preirus (Hitachi Medical Corporation, Tokyo, Japan). A 1–6 MHz microconvex probe or a 2–12 MHz broadband linear probe was used with Arietta60. A 4–8 MHz microconvex probe or a 6–14 MHz broadband linear probe was used with HI VISION Preirus. Long diameter and short diameters of lesions were measured using the US images. When the length of hepatic lesions was not within the echo window, the two diameters were measured using the concurrent CT images. Echo patterns of lesions were classified as solid, cystic, or mixed pattern. Echogenicity in lesions was evaluated as hyperechoic, isoechoic, or hypoechoic compared to that in the surrounding normal liver tissue.

#### 4.3.3. CEUS Using LBs and Sonazoid

First, both LB-CEUS and Sonazoid-CEUS were performed on 13 dogs to compare the US contrast findings for each preparation. Next, LB-CEUS was performed on eight dogs and we analyzed US contrast findings for 21 dogs in total to evaluate diagnostic ability. CEUS was performed with Arietta60 or HI VISION Preirus. A 1–6 MHz microconvex probe or a 2–12 MHz broadband linear probe was used with Arietta60. A 4–9 MHz linear probe was used with HI VISION Preirus. In each US scanner, the mechanical index was set at 0.18–0.22 in contrast harmonic imaging mode. The background gain was adjusted to the minimum level to distinguish the contrast signal from the background tissue signal. Dogs were placed in dorsal recumbency and their cranial abdomen was shaved. If the dogs underwent a concurrent CT examination, they were placed under general anesthesia during CEUS. LBs were administered to dogs intravenously at 0.015 mL/kg (2.5 × 10^9^ particles/mL) through a 22 or 24-G catheter, followed by a 3–5 mL heparinized saline injection. The probe was set on the abdominal region to allow normal liver tissue and focal liver lesions to be scanned on a single image. In order to obtain images of the arterial, portal, and Kupffer phases, real time imaging was continued from 0 to 12 min following drug administration. Every dog was monitored for any acute allergic symptoms occurring due to the administration of LBs. Sonazoid-CEUS was performed on 13 dogs after the disappearance of liver tissue enhancement with LB-CEUS was confirmed. Sonazoid was administered to the dogs intravenously at 0.015 mL/kg (0.85 × 10^9^ particles/mL), and Sonazoid-CEUS was performed in the same way. CEUS was performed by one of three veterinarians (I.Y., Y.M., and K.H.).

#### 4.3.4. Evaluating the Diagnostic Accuracy of LB-CEUS

As previously reported for Sonazoid-CEUS in dogs and humans [19,23,24,25], the timing of three phases for LB-CEUS was defined as follows: arterial phase—defined as the period from LBs administration to the beginning of hepatic portal vein enhancement; portal phase—the period around the peak hepatic portal vein enhancement; Kupffer phase—the period after which hepatic portal vein enhancement completely disappeared. CEUS findings were classified as mentioned below. In the arterial and portal phases, an enhanced pattern of focal liver lesions was classified as a homogenous enhancement pattern (homogenous contrast staining in the whole lesion) or as a heterogenous enhancement pattern (enhancement patterns were mixed within a lesion and it was difficult to classify a single enhancement pattern). Homogenous enhancement patterns were evaluated as hyperenhancing, isoenhancing, or hypoenhancing compared to the surrounding normal liver tissue. In the Kupffer phase, the enhancement patterns of the lesion were evaluated as CD, ID, or ND. When the whole lesion presented an enhancement defect with a clear border to the surrounding normal liver tissue, it was evaluated as CD. In contrast, ID indicated an incomplete enhancement defect with heterogeneous brightness in the lesion. ND indicated that the whole lesion was equally enhanced compared to the surrounding normal liver tissue. Two veterinarians (I.Y. and Y.M.) discussed and evaluated the enhancement patterns of all cases.

### 4.4. Anti-Tumor Effects of LBs Combined with US and Liposomal Doxorubicin

#### 4.4.1. Animals

Three owned dogs were included in this clinical study. The dogs were brought to the Tottori University Veterinary Medical Center between July 2016 and March 2017. It was confirmed that the dogs had spontaneously arising solid tumor on the perianal and cervical region, which were unresectable due to their anatomical locations. It was also confirmed that the dogs did not have serious heart disease, renal dysfunction, and liver dysfunction. The detailed information on the dogs is provided in the results Section 2.3 and Table 3. The clinical protocol for the owned dogs as well as the procedures they underwent was approved by the Ethics Committee at the faculty of Agriculture, Tottori University (ethical approval number: H28-002).

#### 4.4.2. Treatment Protocol

A 22 or 24-G catheter was placed in the cephalic vein of each dog that was sedated and fixed in a suitable position. Hair around the tumor was gently shaved. LBs and liposomal doxorubicin (Doxil, Janssen Pharmaceutical K.K., Tokyo, Japan) were prepared just before use. Doxil (8.0–16.0 mg/m^2^) was administered intravenously for 10–20 min. Following the infusion of Doxil, LBs were administered intravenously at 0.2–0.4 mL/dog (2.5 × 10^9^–5.0 × 10^9^ particles/mL) every 5 min. Simultaneously, therapeutic US irradiation was initiated using a physiotherapeutic US generator (UST-770, ITO Co. Ltd., Tokyo, Japan) and probes with a 15 or 35 mm-diameter circular disk as an US transducer. The output settings for the US irradiation were as follows: frequency of 1 MHz, power intensity of 2 W/cm^2^, and duty cycle (the proportion of the pulse period in which US is generated) of 50% or 100%. US irradiation was performed for 15–40 min. The treatment interval was at least 7 days. The dose of Doxil and LBs, probe, duration of US irradiation, and the number of treatment cycles were determined according to the general condition of each dog.

#### 4.4.3. CEUS for the Assessment of Tumor Vasculature

Immediately before and after each treatment, a tumor lesion was examined using LB-CEUS to determine blood flow in the tumor vasculature as part of theranostics. The US scanners, settings, and probes used here were the same as those described above in the clinical study of CEUS. LBs were administered to the dogs intravenously at 0.015 mL/kg (2.5 × 10^9^ particles/mL), followed by an injection of 3–5 mL heparinized saline. The probe was set on the tumor surface and real-time imaging was performed for 5–10 min.

#### 4.4.4. Measurement of Tumor Size

The length, width, and thickness of each tumor were measured using calipers. Tumor volume was estimated using the formula “length × width × depth × 3.14/6”. To compare the reduction of tumor volume between each dog, relative tumor volume was calculated by dividing tumor volume on day X by tumor volume on day 0.

### 4.5. Statistical Analysis

The differences between the values before the injection of LBs and those 24 h after the injection were analyzed using paired *t-*test in the study on healthy Beagles. The difference between the MGI in LB-CEUS and that in Sonazoid-CEUS was analyzed using Student’s *t-*test in the study of healthy Beagles. The relationship between US contrast findings and lesion malignancy was analyzed with a two-tailed Fisher’s exact test in the clinical study of LB-CEUS. All statistical analyses were performed with a computer software program (XLSTAT, Addinsoft, Paris, France). A *p*-value less than 0.05 was considered significant in all the tests.

## 5. Conclusions

LBs were applied, for the first time, to dogs in the present study; we confirmed the short-term safety and US contrast effect in healthy dogs administered LBs. We also confirmed the usefulness of LB-CEUS in distinguishing malignant lesions from benign lesions in dogs with focal liver lesions. Additionally, we demonstrated that the combination of LBs and US irradiation had the potential to enhance the anti-tumor effect of Doxil in dogs with spontaneously arising tumors. However, further evaluation with a large number of clinical cases is required to confirm the usefulness of LBs because the clinical studies performed here included a small number of dogs. Nevertheless, these results of the present study show the potential of LBs as a theranostic MB preparation in dogs. The present study is expected to be the basis for application to human medicine as a translational research on theranostics.

## Figures and Tables

**Figure 1 cancers-12-02423-f001:**
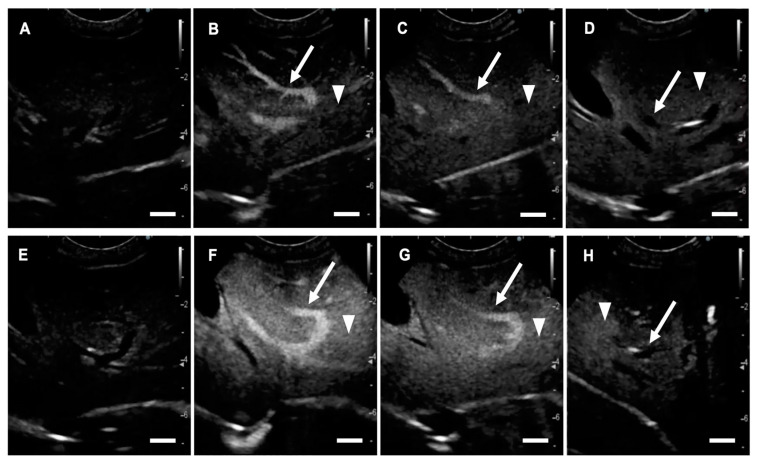
Contrast-enhanced ultrasonography (CEUS) images showing the liver of a healthy Beagle. (**A**–**D**) CEUS using lipid bubbles (LBs); (**E**–**H**) CEUS using Sonazoid; (**A**,**E**) just before; (**B**,**F**) 1 min after; (**C**,**G**) 3 min after; and (**D**,**H**) 15 min after administration of each preparation. (**A**–**D**) The hepatic portal vein (arrows) and liver tissue (arrow heads) are enhanced by the administration of LBs. The enhancement of the hepatic portal vein gradually fades, although liver tissue remains enhanced. (**E**–**H**) The hepatic portal vein (arrows) and liver tissue (arrow heads) are enhanced by the administration of Sonazoid. Scale bar: 1 cm.

**Figure 2 cancers-12-02423-f002:**
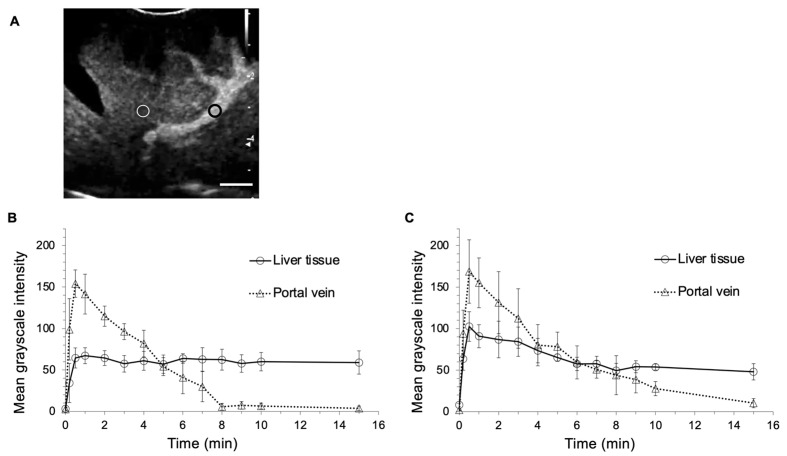
Time-mean grayscale intensity curves of liver tissue and the hepatic portal vein in contrast-enhanced ultrasonography (CEUS). (**A**) A representative image showing the regions of interest on the liver tissue and the hepatic portal vein. A white circle indicates the region of interest on the liver tissue and a black circle indicates the region of interest of the hepatic portal vein. The circles are set on the points at the same level of depth. (**B**) In CEUS using lipid bubbles and (**C**) in CEUS using Sonazoid. Error bars denote the standard deviation (*n* = 6). Scale bar: 1 cm.

**Figure 3 cancers-12-02423-f003:**
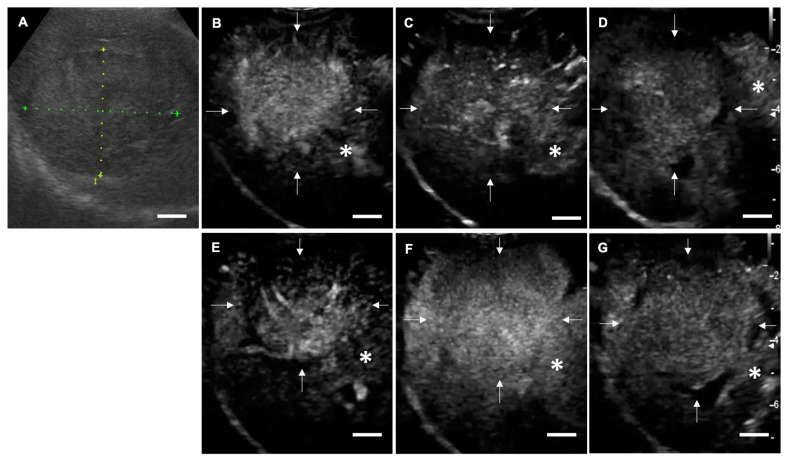
Contrast-enhanced ultrasonography (CEUS) images of hepatocellular adenoma (case No. 1). (**A**) Conventional B mode image. A solid and isoechogenic lesion is observed with a long diameter of 5.97 cm and short diameter of 4.20 cm. (**B**–**D**) Images for CEUS using lipid bubbles (LB-CEUS). (**B**) In the arterial phase, the lesion is hyperenhancing compared to the surrounding liver tissue. (**C**) In the portal phase, the lesion is isoenhancing. (**D**) In the Kupffer phase, no enhancement defect is observed in the lesion. White arrows indicate the lesion margin and asterisks indicate normal liver tissue. (**E**–**G**) Images for CEUS using Sonazoid. (**E**) In the arterial phase, (**F**) the portal phase, and (**G**) the Kupffer phase, the contrast findings are consistent with those for LB-CEUS. Scale bar: 1 cm.

**Figure 4 cancers-12-02423-f004:**
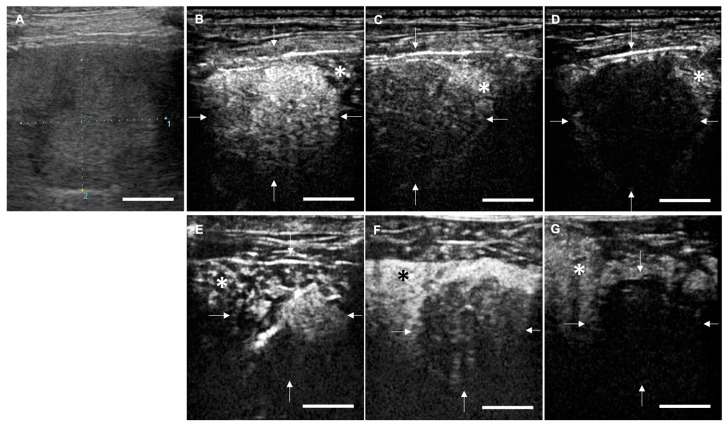
Contrast-enhanced ultrasonography (CEUS) images of hepatocellular carcinoma (case No. 11). (**A**) Conventional B mode image. A solid and isoechogenic lesion is observed with a long diameter of 2.65 cm and short diameter of 2.31 cm. (**B**–**D**) Images for CEUS using lipid bubbles (LB-CEUS). (**B**) In the arterial phase, the lesion is hyperenhancing compared to the surrounding liver tissue. (**C**) In the portal phase, the lesion is hypoenhancing. (**D**) In the Kupffer phase, complete enhancement defect is observed in the lesion. White arrows indicate the lesion margin and asterisks indicate normal liver tissue. (**E**–**G**) Images for CEUS using Sonazoid. (**E**) In the arterial phase, (**F**) the portal phase, and (**G**) the Kupffer phase, the contrast findings are consistent with those for LB-CEUS. Scale bar: 1 cm.

**Figure 5 cancers-12-02423-f005:**
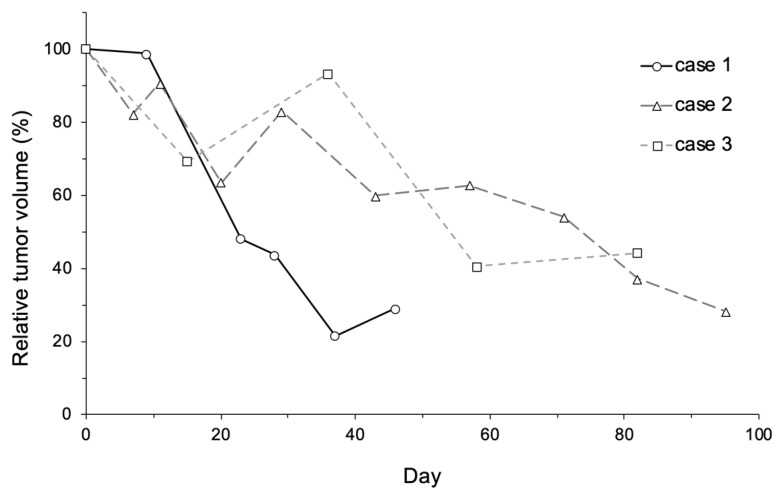
Change in relative tumor volume in each dog. Relative tumor volume was calculated by dividing tumor volume on Day X by tumor volume on Day 0. Relative tumor volume markedly decreased after treatment, to 28.9%, 28.2%, and 44.3% in cases No. 1, 2, and 3, respectively.

**Figure 6 cancers-12-02423-f006:**
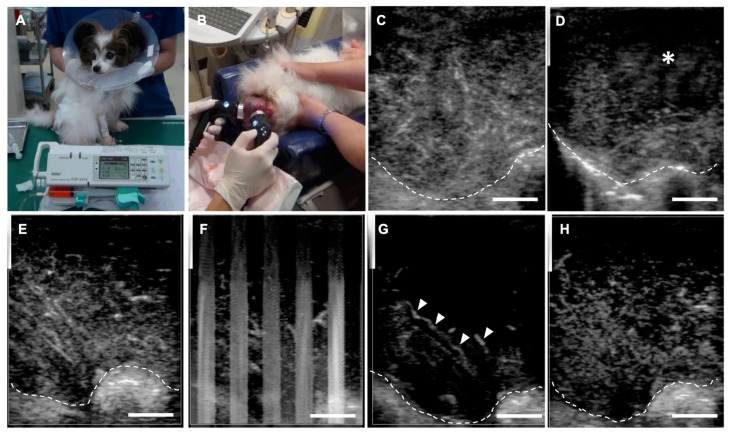
Combination tumor treatment using lipid bubbles (LBs), therapeutic ultrasound (US), and liposomal doxorubicin (Doxil) in case No. 1 with hemangiopericytoma on the perianal region. (**A**) The dog was administered Doxil intravenously at 16 mg/m^2^ for 20 min. (**B**) After infusion of Doxil, the tumor on was irradiated with therapeutic US under sedation. During treatment, LBs were administered intravenously every 5 min. (**C**,**D**) Images of contrast-enhanced ultrasonography using LBs (LB-CEUS) at the first treatment on day 0. LB-CEUS was performed before and after the combination treatment to determine any change in the contrast finding. White dashed lines show the margin of the tumor. (**C**) An image of LB-CEUS just before treatment. The whole tumor is enhanced homogenously 80 s after the injection of LBs. (**D**) After the first treatment, a hypoenhancing area appears in the right-upper part of the tumor tissue (asterisk). (**E**–**H**) LB-CEUS images around the first treatment on day 0. LB-CEUS was performed during the combination treatment to evaluate the influence of therapeutic US on the enhancement in the tumor in real time. (**E**) Before therapeutic US irradiation. The tumor is enhanced homogenously 40 s after the injection of LBs. (**F**) During therapeutic US irradiation. Artifacts induced by therapeutic US irradiation are visualized as white vertical striped patterns. (**G**) At the end of therapeutic US irradiation. Most of the enhancement in the tumor tissue disappears, while LBs reperfusion into the tumor vasculature begins rapidly (arrowheads). (**H**) Twenty seconds after therapeutic US irradiation. LBs are reperfused into the tumor, resulting in the enhancement of the whole tumor tissue again. Scale bar: 1cm.

**Figure 7 cancers-12-02423-f007:**
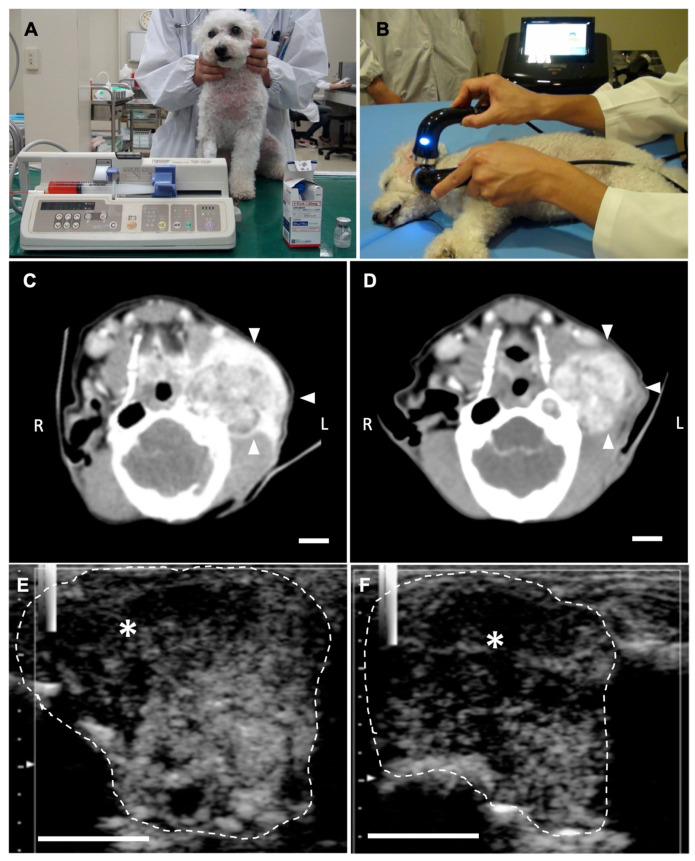
Combination tumor treatment using lipid bubbles (LBs), therapeutic ultrasound (US), and liposomal doxorubicin (Doxil) in case No. 2 with ceruminous gland adenocarcinoma. (**A**) The dog was administered Doxil intravenously at 8–16 mg/m^2^ for 20 min. (**B**) After infusion of Doxil, the tumor at the root of the left ear was irradiated with therapeutic US under sedation. During treatment, LBs were administered intravenously every 5 min. (**C**,**D**) Angiographic computed tomography images for case No. 2. (**C**) Before the first treatment, tumor tissue (arrowheads) was irregularly enhanced by iodine contrast media. The structure of the left ear canal and tympanum is destroyed due to tumor invasion. (**D**) The tumor tissue enhanced by iodine contrast media decreased in size at the third treatment on Day 29. Reconstruction of the left tympanum was also observed. (**E**,**F**) Images of contrast-enhanced ultrasonography using LBs (LB-CEUS) at the third treatment on Day 29. LB-CEUS was performed before and after the combination treatment to determine any change in the contrast finding. White dashed lines show the margin of the tumor. (**E**) An image of LB-CEUS just before treatment. The tumor contains a hypoenhancing area in the left-upper part of the tumor tissue (asterisk). (**F**) An image of LB-CEUS immediately after the treatment. A hypoenhancing area becomes wider in the tumor tissue (asterisk). Scale bar: 1 cm.

**Figure 8 cancers-12-02423-f008:**
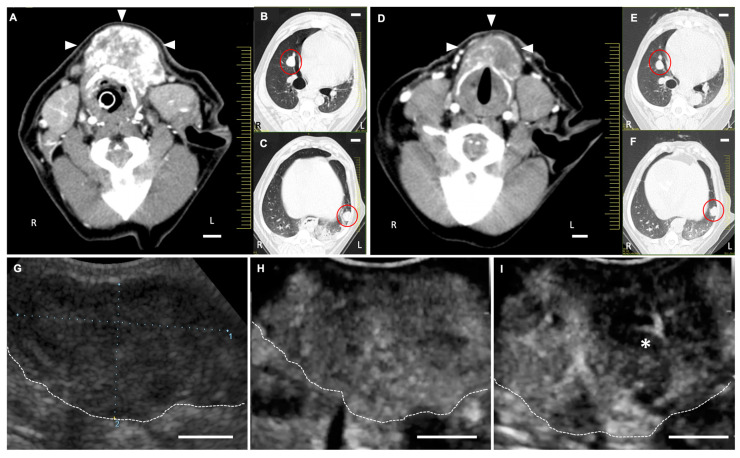
Angiographic computed tomography (CT) and contrast-enhanced ultrasonography using lipid bubbles (LB-CEUS) images in case No. 3. (**A**–**C**) Angiographic CT images on Day 0. (**A**) The dog was clinically diagnosed with suspected thyroid carcinoma. Tumor tissue (arrowheads) is enhanced irregularly by iodine contrast media. Normal thyroid gland is not observed. (**B**,**C**) Pulmonary metastases (red circles) are observed in the chest cavity. (**D**–**F**) Angiographic CT images on Day 58. (**D**) The tumor tissue decreases after the fourth treatment. (**E**,**F**) The pulmonary metastases (red circles) do not change in size. (**G**–**I**) LB-CEUS images of the neck tumor on Day 0. LB-CEUS was performed on tumor tissue before and after tumor treatment to determine any change in the contrast finding. White dashed lines show the margin of the tumor. (**G**) A conventional B mode image shows a hypoechogenic mass in the neck on Day 0. (**H**) An image of LB-CEUS just before the first treatment. The whole tumor is enhanced homogenously 30 s after the injection of LBs. (**I**) Immediately after the first treatment. The enhancement of central area in the tumor (asterisk) decreases. Scale bar: 1 cm.

**Table 1 cancers-12-02423-t001:** Hematological and biochemical tests in healthy Beagles.

Parameters ^1^	Pre-Injection of Lipid Bubbles	Post-Injection (24 h) of Lipid Bubbles	Unit
WBC	106.8 ± 20.1	98.3 ± 17.8	10^2^/μL
RBC	705.0 ± 36.7	728.8 ± 70.5	10^4^/μL
HGB	16.0 ± 1.0	16.5 ± 2.0	g/dL
HCT	44.6 ± 2.9	46.1 ± 5.2	%
PLT *	41.0 ± 9.1	30.7 ± 9.2	10^4^/μL
CRP	0.5 ± 0.4	0.6 ± 0.2	mg/dL
BUN *	15.5 ± 1.8	11.5 ± 0.9	mg/dL
CRE *	0.72 ± 0.1	0.67 ± 0.1	mg/dL
ALT	45.7 ± 5.3	49.5 ± 4.6	U/L
ALP	153.3 ± 39.0	152.5 ± 32.3	U/L
ALB	3.3 ± 0.1	3.1 ± 0.3	g/dL

^1^ WBC, white blood cell; RBC, red blood cell; HGB, hemoglobin; HCT, hematocrit; PLT, platelet; CRP, C-reactive protein; BUN, blood urea nitrogen; CRE, creatinine; ALT, alanine aminotransferase; ALP, alkaline phosphatase; ALB, albumin. * Significant difference (*p* < 0.05) between the value for pre-injection and that for post-injection. All values are shown as mean ± standard deviation (*n* = 6).

**Table 2 cancers-12-02423-t002:** Summary of the results in the clinical study of contrast-enhanced ultrasonography using lipid bubbles and Sonazoid.

Benign/ Malignant	No	Breed	Age (y)	Sex ^1^	Diagnosis	Conventional B Mode			LB-CEUS			Sonazoid-CEUS ^6^		
						Lesion Size ^2^ (cm)	Echo Pattern	Echo-Genicity ^3^	Arterial ^4^ (0–30 s)	Portal ^4^ (30 s–2 min)	Kupffer ^5^ (10 min–)	Arterial ^4^ (0–30 s)	Portal ^4^ (30 s–2 min)	Kupffer ^5^ (10 min–)
Benign	1	Mix	10	M	Hepatocellular adenoma	5.97 × 4.20	solid	iso	hyper	iso	ND	hyper	iso	ND
	2	Chihuahua	12	M	Cholangiocellular adenoma	3.16 × 2.87	mixed	hypo	hetero	hetero	CD	hetero	hetero	CD
	3	Mix	12	SF	Nodular hyperplasia	2.89 × 2.01	solid	iso	hyper	iso	ND	hyper	iso	ND
	4	Welsh Corgi	14	SF	Degeneration and necrosis of liver cells	7.06 × 4.36	solid	iso	iso	iso	ND	iso	iso	ND
	5	Boston Terrier	13	CM	Trichangiectasia	1.04 × 0.92	solid	hyper	iso	iso	ND	iso	iso	ND
	6	Chihuahua	9	SF	Vacuolar degeneration and fatty degeneration of liver cells	1.55 × 1.54	solid	iso	iso	iso	ND	n/a	n/a	n/a
Malignant (primary)	7	Yorkshire Terrier	9	F	Hepatocellular carcinoma	7.68 × 5.46	solid	hypo	hetero	hetero	ID	hetero	hetero	ID
	8	Miniature Dachshund	12	M	Hepatocellular carcinoma	12.02 × 8.49	solid	hypo	hetero	hetero	ID	hetero	hetero	ID
	9	French Bulldog	11	F	Hepatocellular carcinoma	1.56 × 1.12	solid	hypo	iso	hypo	CD	iso	hypo	CD
	10	Chihuahua	11	M	Hepatocellular carcinoma	2.57 × 2.07	solid	hypo	hypo	hypo	CD	hypo	hypo	CD
	11	Mix	11	SF	Hepatocellular carcinoma	2.65 × 2.31	solid	iso	hyper	hypo	CD	hyper	hypo	CD
	12	Miniature Dachshund	14	CM	Hepatocellular carcinoma	4.52 × 4.11	solid	iso	hyper	iso	CD	n/a	n/a	n/a
	13	Golden Retriever	10	CM	Hepatocellular carcinoma	2.92 × 2.58	solid	hypo	hyper	iso	CD	n/a	n/a	n/a
	14	Mix	14	SF	Hepatocellular carcinoma	6.82 *	solid	hypo	hetero	hetero	ID	n/a	n/a	n/a
	15	Beagle	13	F	Hepatocellular carcinoma	12.26 × 11.64	solid	hypo	hetero	hetero	ID	n/a	n/a	n/a
	16	Miniature Schnauzer	12	CM	Cholangiocellular carcinoma	6.05 × 3.76	solid	hypo	hyper	iso	CD	n/a	n/a	n/a
	17	Beagle	8	SF	Epithelial malignant tumor	6.29 × 5.93	solid	hypo	hypo	hypo	CD	n/a	n/a	n/a
Malignant (metastatic)	18	Miniature Dachshund	11	M	Hemangiosarcoma	2.98 × 1.92	solid	iso	hetero	hetero	CD	hetero	hetero	CD
	19	Miniature Schnauzer	9	SF	Malignant melanoma	1.52 × 0.96	solid	hyper	hypo	hypo	CD	hypo	hypo	CD
	20	Jack Russell Terrier	13	CM	Epithelial malignant tumor	9.31 × 5.72	solid	iso	hetero	hetero	ID	hetero	hetero	ID
	21	Miniature Dachshund	11	CM	Nonepithelial malignant tumor	2.14 × 1.40	solid	hypo	hypo	hypo	CD	n/a	n/a	n/a

^1^ M, male; CM, castrated male; F, female; SF, spayed female. ^2^ lesion size, long diameter × short diameter of lesions. ^3^ hyper, hyperechoic; iso, isoechoic, hypo, hypoechoic. ^4^ hyper, hyperenhancing; iso, isoenhancing; hypo, hypoenhancing; hetero, heterogenous enhancement pattern. ^5^ CD, complete enhancement defect; ID, irregular enhancement defect; ND, no enhancement defect. ^6^ n/a, not assessed. * Long diameter could not be measured because it exceeded the length of the echo window.

**Table 3 cancers-12-02423-t003:** Summary of the dogs included in the clinical study.

No.	Breed	Age (y)	Sex ^1^	Weight (kg)	Tumor Type	Tumor Site	Simultaneous Therapy
1	Papillon	14	SF	2.5	Hemangiopericytoma	Perianal	None
2	Toy Poodle	7	F	3.8	Ceruminous gland adenocarcinoma	Left ear	None
3	Welsh Corgi	10	F	10.1	Thyroid carcinoma (suspected)	Thyroid	Toceranib

^1^ F, female; SF, spayed female.

**Table 4 cancers-12-02423-t004:** Information on the treatments for each dog.

No.	US Output Setting				Concentration of Doxil (mg/m^2^)	Number of Treatments	Tumor Volume			Survival Time ^1^ (days)	Outcome
	Frequency (MHz)	Power Intensity (W/cm^2^)	Duty Cycle (%)	Irradiation Time (min)			A: At the First Measurement (cm^3^)	B: At the Last Measurement (cm^3^)	B/A (%)		
1	1	2	50	20–30	16	2	178.0	51.4	28.9	52	Died of a cause unrelated to the tumor
2	1	2	50 or 100	30–40	8–16	6	4.4	1.3	28.2	127	Died of a cause unrelated to the tumor
3	1	2	50	15	15	4	41.0	18.1	44.3	n/a	Lost to follow-up

^1^ n/a, not assessed. Case No. 3 was lost to follow-up by day 83.

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
