# Peer review of "A Pilot Study on Efficacy of Lipid Bubbles for Theranostics in Dogs with Tumors"

_cancers, 2020, doi:10.3390/cancers12092423_

Round 1
Reviewer 1 Report
This is an interesting study conducted by Yokoe et al to evaluate efficacy of lipid bubbles for theranostics in dogs with tumors. Lipid bubbles were applied in the present study to first determine safety and US contrast effect in healthy dogs. Next, they confirmed the usefulness of LB-CEUS in distinguishing malignant lesions from benign lesions in dogs with focal liver lesions. Results suggested that LB-CEUS, as well as Sonazoid-CEUS, could be safe and useful for diagnosing focal liver lesions in dogs. Additionally, they demonstrated that the combination of LBs and US irradiation had the potential to enhance the anti-tumor effect of Doxil in dogs with spontaneously arising tumors. Studies are well conducted, and results are presented appropriately. These studies highlight the importance of theranostics and that LBs have potential for both therapeutic and diagnostic applications in dogs in combination with US irradiation. However, further evaluation with a large number of dogs is required to confirm these studies. Below minor comments need to be addressed.
- Figures 3 and 4 are only representative images of LB-CEUS and Sonazoid-CEUS. Authors should provide the data in the form of table or figure from all the dogs studied.
- Since the number of dogs used to evaluate chemotherapeutics effects are small. More rats should be tested to attain statistical significance. Comment.
Author Response
1. Figures 3 and 4 are only representative images of LB-CEUS and Sonazoid-CEUS. Authors should provide the data in the form of table or figure from all the dogs studied.
Response: We thank the reviewer for the valuable suggention. As instructed, we have included a new table (Table 2), in lieu of the old Table 1, detailing the data for all the 21 dogs used in the clinical study of contrast-enhanced ultrasonography using LBs (page 6, lines 158–161). Moreover, we have added a new subsection “4.3.2. Characterization of focal liver lesions with conventional ultrasonography”in the Materials and Methods (page 17, lines 460–469).
2. Since the number of dogs used to evaluate chemotherapeutics effects are small. More rats should be tested to attain statistical significance. Comment.
Response: We thank the reviewer for this important comment. As has been pointed out, the number of dogs included in the clinical study of the combination treatment using Doxil, LBs, and US was indeed quite small (only three). However, we wish to mention that this treatment was administered an exploratory basis as it has not been used before; therefore, we conducted the clinical study with just three dogs. These three dogs had spontaneously arising tumors and were brought to our veterinary medical center by their owners. As such, it is difficult to include more dogs in the study for determining the statistical significance of results. We consider that the small number of the dogs included in this study is an important limitation to this study, and we have clearly highlighted this in the new paragraph in the Discussion section (page 15, line 371–387). Also, we refer to the limitation again in the Conclusion section (page 19, line 555–557).
Reviewer 2 Report
Abstract
P1 L27: ‘an unresectable solid tumor on the body surface’, body surface is too vague, be more specific.
Introduction
Very-well written and concise.
Materials and Methods
4.1. Preparation of LBs
More details are needed in this section. Molar ratio is not enough. Weights would bed needed. And how was the homogenizing of the lipid dispersion in a perfluoropropane atmosphere achieved? It is currently not enough description for another lab to reproduce this section.
4.3. CEUS in dogs bearing focal liver lesions 352
4.3.1. Animals 353
‘Twenty-one dogs were included in this clinical study’.
Not enough details. A similar assessment as in 4.2.1 is needed: species, gender, age and weight.
‘Dogs with one or more 355 focal liver lesions confirmed with ultrasonography were included’.
How were those lesions acquired? Naturally occurring or induced/ More description is needed.
4.4. Anti-tumor effects of LBs combined with US and liposomal doxorubicin
4.4.1. Animals 397
‘Three dogs with a confirmed inoperable solid tumor on the body surface were included in this clinical study’.
Again these are not rigorous enough.
How was the tumor acquired? Was it induced or naturally occurinfg?
Was the species Beagles again?, what was their age and weight. I guess now I have found this is found in the results. But please consider having those details in the materials and methods too or at least a comment that those details will be found in the results section.
General: Add what happened to all the dogs after the studies were concluded. What was their health afterwards? Was follow-up performed?
A thorough report should be added, even if supplementary data.
Results
P2, L78: ‘No biochemical or hematological changes were observed in healthy Beagles 24 h after LBs 78 administration. In addition, there were no acute allergic symptoms related to LBs administration.’
There is no proof of that nowhere in the manuscript.
Figure 1: How do you explain differences between LB and Sonazoid (this latter which seems better)?
P2 L85: Figure 2 shows the time-mean grayscale intensity (MGI) curve of the hepatic portal vein and 85 liver tissue. Add a representative image of one of the ROIs to understand what it comprises, what are its limits.
P3, L108-119: All those details should also be found in the materials and methods
P4, L124: ‘findings of each phase in LB-CEUS and Sonazoid-CEUS were consistent in 13 dogs’. It is unclear which dogs were uses for LB-CEUS and which ones were used for Sonazoid-CEUS. Also, Table 1 seems to show only LB-CEUS.
P7, L163: ‘The survival time or follow-up period was 52, 127, and 83 days in cases No. 163 1, 2, and 3, respectively’. Well, be precise, which was survival time and which was the follow-up period. It looks like two different variables that shouldn’t be pulled together? Also does that suggest that some dogs eventually died? More clarity on that is needed.
Discussion
It is unclear why Sonazoid or other commercial MBs could not be used for the purpose of drug delivery. It is understood that the Sonazoid have not been optimized for application in therapeutics or theranostics but it may then be worth to investigate their potential, rather than the LBs. Clearly the results show indeed that LB-CEUS is not as good as Sonazoid-CEUS, which questions whether this is a good enough, and the authors state whether this difference has an influence on contrast findings is necessary in clinical use, yet they do not show it. What is also missing is a comparison between the LB vs Sonazoid system on animals with similar tumors to assess whether the LB is in fact superior for drug delivery.
Furthermore, there is no mechanisms of action depicted in the manuscript, so the studies are rather descriptive.
Author Response
Comments and Suggestions for Authors
Abstract
P1 L27: ‘an unresectable solid tumor on the body surface’, body surface is too vague, be more specific.
Response: We thank the reviewer for this reasonable comment. The phrase “an unresectable solid tumor on the body surface” has been replaced with “an anatomically unresectable solid tumor on the perianal and cervical region” (the Abstract section, page 1, lines 27–28). We have included the details about the tumor locations and the reason why they were unresectable in the Results section (2.3. Anti-tumor effects of LBs in combination with therapeutic US, and liposomal doxorubicin, page 8, lines 183–186).
Introduction
Very-well written and concise.
Response: We thank the reviewer for this encouraging comment.
Materials and Methods
4.1. Preparation of LBs
More details are needed in this section. Molar ratio is not enough. Weights would bed needed. And how was the homogenizing of the lipid dispersion in a perfluoropropane atmosphere achieved? It is currently not enough description for another lab to reproduce this section.
Response: We thank the reviewer for this pertinent comment. For the benefit of other researchers who might be interested in performing the procedure, we have provided the relevant details in the revised manuscript, as suggested by the reviewer (Materials and Methods section, “4.1. Preparation of LBs,” page 16, lines 391–416).
4.3. CEUS in dogs bearing focal liver lesions 352
4.3.1. Animals 353
‘Twenty-one dogs were included in this clinical study’.
Not enough details. A similar assessment as in 4.2.1 is needed: species, gender, age and weight.
‘Dogs with one or more 355 focal liver lesions confirmed with ultrasonography were included’.
How were those lesions acquired? Naturally occurring or induced/ More description is needed.
Response: We thank the reviewer for the specific instructions. As instructed, we have included a new table (Table 2), in lieu of the old Table 1, detailing the data for all the 21 dogs used in the clinical study of contrast-enhanced ultrasonography using LBs (page 6, lines 158–161). The animals included in the clinical study were owned dogs with naturally-occurring hepatic lesions. We clarified that in the Materials & Methods section (“4.3.1. Animals,” page 17, lines 453–455). We have added sentences to provide information on the occurrence of lesions and have referred the readers to the following section for detailed information: Materials and Methods section:“4.3.1. Animals” (page 17, line 452–453).
4.4. Anti-tumor effects of LBs combined with US and liposomal doxorubicin
4.4.1. Animals 397
‘Three dogs with a confirmed inoperable solid tumor on the body surface were included in this clinical study’.
Again these are not rigorous enough.
How was the tumor acquired? Was it induced or naturally occurinfg?
Was the species Beagles again?, what was their age and weight. I guess now I have found this is found in the results. But please consider having those details in the materials and methods too or at least a comment that those details will be found in the results section.
General: Add what happened to all the dogs after the studies were concluded. What was their health afterwards? Was follow-up performed?
A thorough report should be added, even if supplementary data.
Response: We again thank the reviewer for these specific instructions. As in the clinical study of CEUS, owned dogs with spontaneously arising tumors were included in this clinical study. We have added sentences to refer the readers to the following section for detailed information: Materials and Methods section: “4.4.1. Animals” (page 18, lines 509–510). In addition, the outcomes of the three dogs were added in the Results section (“2.3. Anti-tumor effects of LBs in combination with therapeutic US, and liposomal doxorubicin;” page 9, lines 189–191) and in Table 4 (page 10, lines 198–199). Furthermore, we have additionally submitted a supplementary material and described the clinical courses of the three cases.
Results
P2, L78: ‘No biochemical or hematological changes were observed in healthy Beagles 24 h after LBs 78 administration. In addition, there were no acute allergic symptoms related to LBs administration.’
There is no proof of that nowhere in the manuscript.
Response: We appreciate the pertinent comment made by the reviewer. A new table (Table 1), providing the results of hematological and biochemical tests has been included in the revised manuscript (page 3, lines 110–114). We have also modified the relavant sentence in the Abstract and the Results section (“Abstract, ” page 1, line 24, and “2.1. US contrast effect and safety of LBs in healthy Beagles”, page 2, lines 79–84). In addition, the concrete symptoms observed through general monitoring have been described in section 2.1 (“2.1. US contrast effect and safety of LBs in healthy Beagles,” page 2, line 84–87). In the Discussion section, the description of the safety of LBs was appropriately rephrased for clarity (page 13, lines 270–278). Moreover, in the Materials & Methods section, “4.5. Statistical analysis,”a sentence about the methods used for the analysis of blood tests has been added (page 19, lines 543–544). We has rephrased the sentences in the result and discussion sections related to the safety of LBs in clinical study for CEUS (page 5, line 143, page 14, lines 328–329, and page 15, line 332).
Figure 1: How do you explain differences between LB and Sonazoid (this latter which seems better)?
Response: We thank the reviewer for this pertinent query. In the results of the Beagle study, the peak MGI value of liver tissue enhancement was higher in Sonazoid-CEUS. However, we do not consider that it was an important difference. It was revealed that both of LBs and Sonazoid had the three phases of the arterial, portal, and Kupffer phase. Therefore, we consider that LBs can be used for CEUS equally to Sonazoid. We would like to inform the reviewer that we have rephrased the legend for clarity (Figure 1, pages 4 , lines 116–122). Moreover, we modified the sentence of the Result and Discussion section to make our statement obvious (page 2, lines 99–103, and page 14, lines 286–303).
P2 L85: Figure 2 shows the time-mean grayscale intensity (MGI) curve of the hepatic portal vein and 85 liver tissue. Add a representative image of one of the ROIs to understand what it comprises, what are its limits.
Response: We thank the reviewer for this valuable suggestion. As requested, we have added another image to present the ROIs clearly as Figure 2A (page 4, lines 123–129).
P3, L108-119: All those details should also be found in the materials and methods
Response: We appreciate the reviewer’s concern for inclusion of the relavant details. To avoid redundancy, we have added a new sentence, referring the readers to the Results section for detailed information (Materials and Methods section, “4.3.1. Animals,” page 17, lines 453–454).
P4, L124: ‘findings of each phase in LB-CEUS and Sonazoid-CEUS were consistent in 13 dogs’. It is unclear which dogs were uses for LB-CEUS and which ones were used for Sonazoid-CEUS. Also, Table 1 seems to show only LB-CEUS.
Response: We thank the reviewer for this reasonable comment. We have included a new table summarizing the findings in each CEUS using LBs and Sonazoid (Table 2, page 6, line 158–161).
P7, L163: ‘The survival time or follow-up period was 52, 127, and 83 days in cases No. 163 1, 2, and 3, respectively’. Well, be precise, which was survival time and which was the follow-up period. It looks like two different variables that shouldn’t be pulled together? Also does that suggest that some dogs eventually died? More clarity on that is needed.
Response: We thank the reviewer for this pertinent comment. As suggested, we have clearly described the survival time and the follow-up period in a new sentence added in the Results section (“2.3. Anti-tumor effects of LBs in combination with therapeutic US, and liposomal doxorubicin,” page 8, line 189–191). In addition, a new row for “outcome” has been added to the Table 4 (page 9, lines 198–199).
Discussion
It is unclear why Sonazoid or other commercial MBs could not be used for the purpose of drug delivery. It is understood that the Sonazoid have not been optimized for application in therapeutics or theranostics but it may then be worth to investigate their potential, rather than the LBs.
Response: We appreciate the reviewer’s comment on this point. To our knowledge, there are only a few preclinical studies on drug delivery using Sonazoid (Sasaki et al., 2013, Sasaki et al., 2017); therefore, the evidence for initiating a clinical study is considered insufficient. On the contrary, several studies on drug delivery using LBs and US have been performed in vitro and in vivo. There are reports on drug delivery of pDNA (Suzuki et al., 2007, Suzuki et al., 2008), drug delivery of conventional doxorubicin into tumor tissue (Ueno et al., 2011), drug delivery of high molecular weight agent into brain (Omata et al., 2019), and enhancement of vascular permeability (Unga et al., 2019). Therefore, we assumed that LBs would be a promising reagent for therapeutic use, conceptualizing the present study. Currently, there are no commercial microbubbles except for Sonazoid in Japan; we have been making attempts to develop LBs as a new microbubble preparation. We do understand the reviewer’s concern; however, we would like the reviewers to consider the evaluation of the application of LBs for therapeutics and theranostics in large animals for the first time.
Clearly the results show indeed that LB-CEUS is not as good as Sonazoid-CEUS, which questions whether this is a good enough, and the authors state whether this difference has an influence on contrast findings is necessary in clinical use, yet they do not show it.
Response: We appreciate the reviewer’s comment on this point. If we understand correctly, by “the results,” the reviewer means the results for LB-CEUS in dogs with focal liver lesions, and the reviewer is stating that the diagnostic value of LB-CEUS was not as good as Sonazoid-CEUS. If this is the case, we wish to mention that our results indicate a good performance, as previously reported, for Sonazoid-CEUS. If the reviewer is referring to the results for the Beagles (Figure 1 and 2), we have presented our interpretation of the results above. It is considered that the superiority of US contrast agents should not be evaluated in time-MGI curve or in individual images in the Beagles. If our interpretation of the reviewer’s comment is wrong, we sincerely apologize and would appreciate further clarification of the point being made by the reviewer.
What is also missing is a comparison between the LB vs Sonazoid system on animals with similar tumors to assess whether the LB is in fact superior for drug delivery.
Response: We appreciate the reviewer’s comment. However, in the present study, we performed the clinical study only using the LBs system. Therefore, we cannot comment on the superiority of LBs over Sonazoid for drug delivery. In the future, we would like to evaluate the differences in in vivo drug delivery by LBs and Sonazoid.
Furthermore, there is no mechanisms of action depicted in the manuscript, so the studies are rather descriptive.
Response: We appreciate the reviewer’s comment on this point. As pointed out by the reviewer, the present study appears descriptive because none of the experiments in this study could explain the mechanism of actions underlying the US contrast effects and drug delivery effects. However, we have described possible mechanisms of actions in the Discussion section (page 14, lines 321–323 and page 15, lines 358–369). Nonetheless, the mechanisms described in the manuscript are only presumptive, and we have, therefore, highlighted this fact as a limitation of this study (the Discussion section, page 15, lines 369–370, and page 15–16, lines 371–387).
Reviewer 3 Report
I would like to recommend the publication of this work without reservation. This translational research on large animals bridges the mouse study and human clinical study on the theranostic effects of LBs and US irradiation. The study is carefully designed and well performed. The manuscript is well written and organized.
Author Response
Comments and Suggestions for Authors
I would like to recommend the publication of this work without reservation. This translational research on large animals bridges the mouse study and human clinical study on the theranostic effects of LBs and US irradiation. The study is carefully designed and well performed. The manuscript is well written and organized.
Response: We thank the reviewer for appreciating our work and for the thoughtful comment.
Reviewer 4 Report
In this manuscript, the authors demonstrate the efficacy of LB-CEUS and Doxil therapy in canine tumors. They also characterized and compared their LB-CEUS strategy with Sonazoid in canine liver lesions, to identify the different enhancement phases. It is a well-written manuscript, with some significant therapeutic efficacy results, albeit in a small number of dogs. I think it can be accepted for publication after addressing the following minor concerns:
1) Please include and discuss the following important points in the Discussion:
a) Although biochemical and hematological parameters were checked. Sonazoid is known to cause heart rate irregularities. Did the authors notice these in their dogs? Also, what are the implications of using perfluoropropane as opposed to perfluorobutane used in Sonazoid?
b) For the first and second studies, the dogs are of various ages. Did the authors notice any inherent differences in uptake and enhancement in the older dogs?
c) Case no 3. is interesting because it was also receiving toceranib, but the tumor reduced to 44%. Compared to the other 2 cases, this was still lower. Is it because this is thyroid carcinoma, or is it because it had metastasized?
d) Fig 5: Case no. 1 seems like it had the fastest response. Does that imply that this theranostic strategy can vary between different tumors? What could be the reason for the upward and downward trend in tumor growth for Case 2 and 3?
e) Although the survival times/follow-up periods are given for the 2 cases, it will be helpful to clarify which they were. Did the treatment increase survival or did the followup not last till survival?
3) For Fig 2, I suggest making the lines thicker or changing the legend in some way, to enable sharper contrast between the 2 lines for easy identification.
4) Table 1: I suggest putting units (sec). Also, for complete transparency, please also include the data for Sonazoid.
5) Fig 3 and 4: Please incorporate the sizes of the focal liver lesions. Are there size-dependent differences in uptake/enhancement ?
6) Please include images for Case 2 and US images for Case 1.
7) Line 289: should be changed to "primary tumor tissue" considering no effect was observed on lung mets.
Author Response
In this manuscript, the authors demonstrate the efficacy of LB-CEUS and Doxil therapy in canine tumors. They also characterized and compared their LB-CEUS strategy with Sonazoid in canine liver lesions, to identify the different enhancement phases. It is a well-written manuscript, with some significant therapeutic efficacy results, albeit in a small number of dogs. I think it can be accepted for publication after addressing the following minor concerns:
1) Please include and discuss the following important points in the Discussion:
- a) Although biochemical and hematological parameters were checked. Sonazoid is known to cause heart rate irregularities. Did the authors notice these in their dogs? Also, what are the implications of using perfluoropropane as opposed to perfluorobutane used in Sonazoid?
Response: We appreciate the comment made by the reviewer. In the present study, we did not check the changes in the heart rate as a result of LBs and Sonazoid injection. According to our knowledge, heart rate irregularity due to Sonazoid-CEUS for humans was reported by Yi-Hong in 2019. In the report, 3 of 54 patients (5.6%) had heart rate irregularity, which was asymptomatic or mild symptomatic. We are afraid that we could not find any other report referring to a relation between irregularity in the heart rate and Sonazoid injection not only in humans, as well as in dogs. Therefore, we considered that it was not essential to monitor the heart rate in the dogs.
Perfluoropropane is contained in the other commercial microbubbles, such as Definity and Optison, which are available in USA. In addition, perfluoropropane has been used for ophthalmological applications in Japan. These facts imply that pefluorepropane is considered a safe gas for the human body. Conversely, it was difficult for us to obtain perflubutane (Sonazoid gas core) because we need to import it from the USA. Therefore, why we chose perfluoropropane as a LBs gas core.
- b) For the first and second studies, the dogs are of various ages. Did the authors notice any inherent differences in uptake and enhancement in the older dogs?
Response: We appreciate the query posed by the reviewer. As mentioned, the ages of the dogs were different in the CEUS studies in Beagles and owned dogs. Although we did not evaluate the differences in the uptake or enhancement of LBs, we believe that ages had little influence on these factors. We rather thought that the hemodynamics of animals (without/under anesthesia, hyper/hypotension, and so on) had an influence on the beginning or the duration time of the arterial, portal, and Kupffer phases. However, we did not refer to these considerations because they were not evaluated quantitatively in the present study.
- c) Case no 3. is interesting because it was also receiving toceranib, but the tumor reduced to 44%. Compared to the other 2 cases, this was still lower. Is it because this is thyroid carcinoma, or is it because it had metastasized?
Response: We appreciate the reviewer’s comment on this point. In case No. 3, we used toceranib for simultaneous therapy because it had lung metastases. As a result, lung metastases did not change in size; therefore, we thought that toceranib had some effect in suppressing the progression of metastases. As pointed out, the reason why the anti-tumor effect in case No. 3 was relatively at low level is unknown. However, we believe that the differences between case No. 3 and the other dogs were not very important because the dogs did not die from the progression of the tumor.
- d) Fig 5: Case no. 1 seems like it had the fastest response. Does that imply that this theranostic strategy can vary between different tumors? What could be the reason for the upward and downward trend in tumor growth for Case 2 and 3?
Response: We thank the reviewer for the important comment. We believe that the tumor type is an important factor for the anti-tumor effect of this combined treatment. Similarly, characterization of the tumor, such as in terms of hardness, location, and microvessel density is also important. In addition, the combination of LBs and US is considered to enhance the efficacy of Doxil in this treatment. Therefore, the degree of Doxil distribution in the tumor tissue and US penetration can influence the treatment efficacy. Case No. 1 had a hemangiopericytoma in the perianal region. The tumor was relatively soft and might have had the fastest response. However, this is just a presumption, and we have not referred to it in the manuscript.
We considered that upward and downward trends in case No. 2 were caused by secondary inflammation around the tumor tissue due to the treatments. In case No. 3, there might have been an artificial error because the tumors were measured with calipers. Slight errors in the order of millimeters can have some influence on the calculated volume.
- e) Although the survival times/follow-up periods are given for the 2 cases, it will be helpful to clarify which they were. Did the treatment increase survival or did the followup not last till survival?
Response: We appreciate the reviewer for the valuable suggestion. As suggested, we have clearly discussed the survival time and the follow-up period in the newly added sentence in the Results section (2.3. Anti-tumor effects of LBs in combination with therapeutic US, and liposomal doxorubicin, page 8, lines 189–191). In addition, a new row for “outcome” has been added to the Table 4 (page 9, lines 198–199).
3) For Fig 2, I suggest making the lines thicker or changing the legend in some way, to enable sharper contrast between the 2 lines for easy identification.
Response: We appreciate the reviewer’s suggestion. Figure 2 was modified for ease of identification (page 4, lines 123–129).
4) Table 1: I suggest putting units (sec). Also, for complete transparency, please also include the data for Sonazoid.
Response: We thank the reviewer for this comment. The old Table 1 has been removed, and new Table 2 detailing data for all the 21 dogs used in the clinical study of contrast-enhanced ultrasonography using LBs have been included. The unit (second) has been mentioned in the new Table 2 (page 6, lines 158–161).
5) Fig 3 and 4: Please incorporate the sizes of the focal liver lesions. Are there size-dependent differences in uptake/enhancement ?
Response: We appreciate the reviewer’s comment. Tumor sizes for all the dogs have been mentioned in the new Table 2 (page 6, lines 158–161), as suggested by the reviewer. In the legends of Figure 3 and 4, the tumor size has been added (page 7, lines 164–165; page 7–8, lines 173–174). In relatively large malignant lesions, the enhancement pattern was likely to be hetero enhancement. However, we did not mention it in the manuscript because it was not evaluated quantitatively.
6) Please include images for Case 2 and US images for Case 1.
Response: We appreciate the reviewer’s comment on this point. As suggested, we have included the US images for case No. 1 in the new Figure 6 (page 11, lines 216–234). Moreover, the relevant sentence in the Results section has been appropriately modified (2.3. Anti-tumor effects of LBs in combination with therapeutic US, and liposomal doxorubicin, page 11, lines 204–207). Images for case No. 2 have been presented in the new Figure 7. In Figure 7E and 7F, we have added the US images for the tumor tissue (page 12, lines 237–251).
7) Line 289: should be changed to "primary tumor tissue" considering no effect was observed on lung mets.
Response: We thank the reviewer for this comment. “the cervical mass” has been changed to “the primary tumor tissue in the cervical region” (Discussion section, page 15, lines 347–348).